# Ambient Environmental Parameter Estimation for Reliable Diffusive Molecular Communications

**Shota Toriyama [1], Shoma Hasegawa [1], Jens Kirchner [2], Georg Fischer [2] and Daisuke Anzai [1,*]**

[1] Graduate School of Engineering, Nagoya Institute of Technology, Nagoya 466-8555, Japan
[2] Institute for Electronics Engineering, Friedrich-Alexander-Universität Erlangen-Nürnberg (FAU), 91058 Erlangen, Germany
* Correspondence: anzai@nitech.ac.jp; Tel.: +81-52-735-5389

**Abstract:** Molecular communication is a promising communication technology that uses biomolecules such as proteins and ions to establish a communication link between nanoscale devices. In diffusive molecular communication, which uses diffusion characteristics of transfer molecules, the diffusion mechanism is mathematically derived as a Channel Impulse Response (CIR) to design an optimal detector structure. However, an ideal environment is assumed for deriving a CIR. Hence there is a concern that developed systems based on the derived CIR may not operate well in a realistic environment. In this study, based on the finite element method (FEM), we constructed a model of the environment with heterogeneous temperature distribution and actual volume of transmitting molecules to not only demodulate the bit information via maximum likelihood sequence estimation (MLSE) but also to estimate the temperature and volume of the transmitting molecules. Furthermore, in this study, we evaluated the performance of the MLSE method and investigated the effects of ambient environmental temperature distribution and volume of the transmitted molecules on diffusive molecular communication. The evaluation results demonstrated that the proposed method can improve the communication performance by approximately 9 dB by estimating the temperature and transmit molecule volume.

**Keywords:** diffusive molecular communications; MLSE; ambient environmental parameter estimation

## 1. Introduction

In recent years, research on molecular communication has been more active with the expectation of realizing nano-networks [1–3]. Molecular communication (MC) is a communication technology that uses molecules such as proteins and ions to transmit information. MC differs from existing communication technologies, which use electromagnetic waves, because it uses biomolecules for information transmission [4,5]. Most molecular communications are a few nanometers to a few micrometers in size and exhibit biocompatibility. These characteristics of MC are expected to lead to applications in environments, including medical and healthcare fields, where existing communication technologies are not applicable [6,7]. An example of the application of MC is the communication between bionanomachines [8]. Given that it is difficult to miniaturize existing wireless communication technologies because the length of the antenna is dependent on wavelength, MC can potentially be applied for bionanomachines. Furthermore, the application of bionanomachines in drug delivery systems (DDS) is under consideration. Molecular communication is expected to be applied not only to the body, but also to the out-of-body environment. An example involves the use of MC to transport materials and reagents between components, such as sensors and reactors in lab-on-chip [7].

In recent years, not only short-range (a few nanometers to a few micrometers) communication, but also long-range (centimeter-order) MC methods have been proposed wherein blood is used as a medium to carry information molecules and transmit information [9]. Furthermore, the characteristics of a micro-fluidic network have been characterized, and a

dielectrophoretic relay-assisted MC system has been modeled using transmission line technology, as an example, for establishing communication between various entities in lab-on-chip [10]. Moreover, other methods, such as the Monte Carlo method, simplified Poisson method, simplified Gaussian method, and simplified gamma method, have been proposed to study the inter-sign interference problem [11]. Additionally, recent studies have experimentally evaluated pH-based coding systems [12,13] and the use of superparamagnetic iron oxide nanoparticles (SPIONs) as transmission particles [14,15].

One form of molecular communication is diffusive molecular communication, which is a type of communication based on the diffusion of released information transfer molecules [16]. In recent years, research on diffusive molecular communication has led to the development of a system with multiple-input multiple-output (MIMO) [17,18] channels, and a neural network detection method based on sequential learning [19]. Additionally, a mathematical model was proposed to investigate the effect of multiple measurements of molecular concentration on the performance of mobile molecular communication [20]. In diffusive molecular communication, the diffusion model of a molecule is mathematically modeled as channel impulse response (CIR). This enables the design of optimal detectors and other devices based on CIR. In previous studies, CIR has been derived for diffusive molecular communication, and communication methods wherein thresholds based on the CIR have been examined [16,21]. However, a derived CIR assumes an idealized environment, where the temperature is constant during diffusion and the transmitting molecule is a point-wave source to simplify the boundary conditions. This assumption does not reflect the characteristics of real-world molecular communication.

In this study, we assumed the application of molecular communication in a lab-on-chip as a change in temperature. We analyzed over time in the number of received molecules when environmental temperature was varied from 10 to 40 °C, and we focused on 25 and 35 °C to evaluate bit error rate (BER) characteristics because the temperature on the lab-on-chip increases from room temperature by 11 °C on the chip [22]. Furthermore, in diffusive molecular communication, most system models assume that the transmitting molecule is a point source. Assuming the transmitting molecule as a point source and simplifying the system model, it is possible to solve the diffusion equation in closed form. However, since the transmitting molecule has a volume in the real environment, the diffusion equation solved under the assumption of point source is no longer valid, so it is necessary to consider the volume of the transmitting molecule.

## 2. System Model of Diffusive Molecular Communications

### 2.1. System Overview

Figure 1 shows the block diagram of diffusive molecular communication. In diffusive molecular communication, a signal is converted into a biochemical phenomenon by a modulator and a transmitting molecule in response to an external input. The modulated signal propagates through the molecular communication channel, is received by the receiving molecule, and is demodulated to transfer information. Figure 2 illustrates the molecular communication system model. The system model assumes a three-dimensional fluid environment in the infinite region, and the temperature and viscosity in the environment are constant. Additionally, it is assumed that the transmitting molecule is a point source with no volume at position $r_0 = (x_0, y_0, z_0)$. The receiving molecule is assumed as a sphere with the center coordinate at the origin and a radius of $a$. It is assumed that green fluorescent protein (GFP) is used as the receiving nanomachine, and the amount of light is measured via a photodetector. Multiple circular receptors with radius $r_s$ exist on the surface of the receiving molecule, and the number of these receptors is $M$. By assuming one type of molecule as a signal transduction molecule, the molecule is denoted as molecule A. Note that the receptor molecules distributed randomly on the surface, where the fluctuations of the receiver molecules should be, affected the connection of transmitter and receiver molecules. The modulation is performed by On-Off Keying (OOK). When the transmission bit is 1, molecule A is released instantaneously at the start time of the symbol interval,

and when the transmission bit is 0, it does not emit. After the signal transduction molecule is released into the environment, it diffuses in all directions via Brownian motion with a diffusion coefficient $D_A$. At that time, it is assumed that molecule A follows the decomposition reaction as

$$A \xrightarrow{k_d} \phi \qquad (1)$$

where $k_d$ is the decomposition reaction coefficient. Furthermore, $\phi$ indicates chemical species that do not contribute to communication. After molecule A has been released from the transmitting molecule, it reaches the surface of the receiving molecule via diffusion and reacts with molecule B, which is located on the receptor on the surface of the receiving molecule. The product at this time is referred to as molecule C. The chemical reaction at the receptor can be written by

$$A + B \underset{k_b}{\overset{k_f}{\rightleftharpoons}} C \qquad (2)$$

where $k_f$ and $k_b$ denote the reaction rate constants of the forward and reverse reactions, respectively.

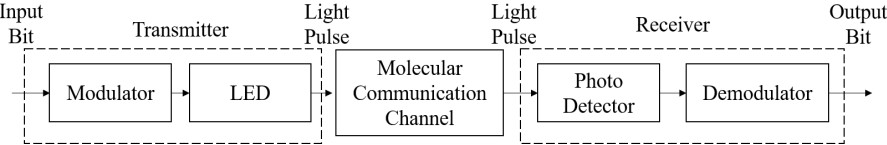

**Figure 1.** Block diagram of diffusive molecular communication.

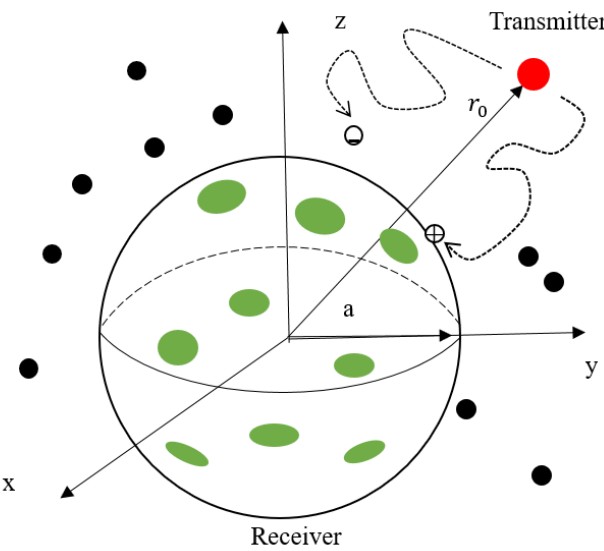

● Signaling molecules A        ⊕ Activated receptor molecules C

● Receiver surface molecules B        ⊖ Degraded molecules $\in \{\emptyset\}$

**Figure 2.** System model of diffusive molecular communications.

### 2.2. CIR for Diffusive Molecular Communication

$P_{AC}(r, t|r_0)$ is the probability density function that molecule *A* emitted from the transmitting molecule at position $r_0$ at time $t = 0$ received as molecule *C* by the receptor on the surface of the receiver molecule. Then, the probability density function that the released molecule *A* exists at position *r* at time *t* is $P_A(r, t|r_0)$. Hence, the diffusion equation in

diffusive molecular communication is expressed by the following equation as the diffusion reaction equation, including the decomposition reaction.

$$\frac{\partial P_A(r,t|r_0)}{\partial t} = D_A \nabla^2 P_A(r,t|r_0) - k_{\mathrm{d}} P_A(r,t|r_0) \tag{3}$$

Here, $\nabla^2$ is the Laplacian in spherical coordinates. Furthermore, (4) is applied as the initial condition.

$$P_A(r, t \to 0 | r_0) = \frac{1}{4\pi r_0^2} \delta(r - r_0) \tag{4}$$

In the above equation, $\delta$ denotes the delta function. Number (4) indicates that the molecule is released in impulses from the transmitting molecule. Furthermore, applying the boundary conditions, we obtain (5) and (6) as

$$\lim_{r \to \infty} P_A(r,t|r_0) = 0 \tag{5}$$

$$4\pi a^2 D_A \left. \frac{\partial P_A(r,t|r_0)}{\partial r} \right|_{r=a} = k_{\mathbf{f}}^\star P_A(a,t|r_0) - k_{\mathrm{b}} \int_0^t 4\pi a^2 D_A \left. \frac{\partial P_A(r,\tau|r_0)}{\partial r} \right|_{r=a} d\tau \tag{6}$$

where $k_{\mathbf{f}}^\star$ is the corrected reaction rate coefficient in the positive direction. the reception probability on the surface of the receiving molecule is expressed as

$$-J(\mathbf{r},t|\mathbf{r_0})|_{\mathbf{r} \in \Omega} = D_A \nabla P_A(\mathbf{r},t|\mathbf{r_0})|_{\mathbf{r} \in \Omega} \tag{7}$$

where $\Omega$ denotes the receptor region on the surface of the receiving molecule. Molecules $A$, which are instantaneously released from the transmitting molecule, are observed as molecules $C$ after reacting with molecules $B$ on the surface of the receiving molecules. The number of molecules observed by the receiving molecule is given by

$$N_C(t|r_0) = N_A P_{AC}(t|r_0) \tag{8}$$

where $N_A$ is the number of molecules $A$ released.

*2.3. Effect of Temperature Change on CIR*

　　Figures 3 and 4 show system models for homogeneous and heterogeneous temperature distribution, respectively. In the heterogeneous environment (Figure 4), the temperatures in the left and right regions differ. The diffusion coefficient and reaction rate coefficient are affected by changes in the temperature. Based on the Stokes-Einstein relation, the diffusion coefficient is obtained as

$$D = \frac{C_B T}{6\pi\eta R_{\mathrm{A}}} \tag{9}$$

where $C_B$ denotes the Boltzmann constant, $T$ is the absolute temperature, $\eta$ is the viscosity of the fluid environment, and $R_A$ is the Brownian particle radius. Conversely, the reaction rate coefficient is expressed by the Arrhenius equation as follows:

$$k = A_k \exp\left(-\frac{E_a}{RT}\right) \tag{10}$$

Here, $A_k$ denotes a frequency factor and corresponds to the collision frequency of the chemical species at temperature $T$. Furthermore, $E_a$ denotes the activation energy, which corresponds to the threshold energy of the reaction, and $R$ denotes the gas constant. It is noted that the collision frequency $A_k$ can be estimated by the environment parameters, such as temperature and activation entropy.

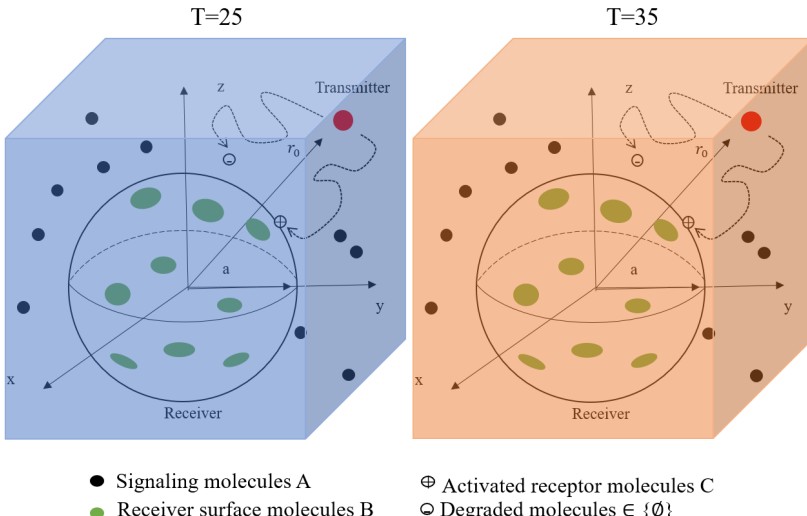

**Figure 3.** System model with homogeneous temperature distribution.

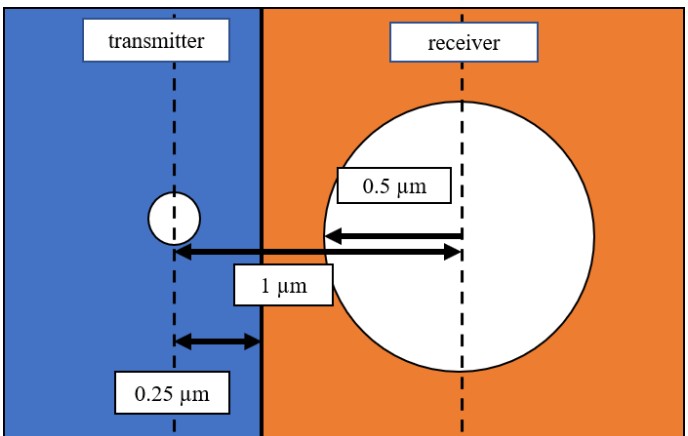

**Figure 4.** System model with heterogeneous temperature distribution.

### 2.4. Effect of Transmitter Molecule Volume on CIR

Figure 5 shows the system model with the transmitter having a finite volume. It is assumed that molecule $A$ is released from the surface of a spherical transmitting molecule with radius $a_{TX}$. Hence, it is considered as homogeneous regardless of the position on the surface of the transmitting molecule, and the emission amount is the same as that in the case of a point source. Given that molecule $A$ is released instantaneously at the start time of the symbol section, it is necessary to change the initial condition (4) of the diffusion equation as follows:

$$
P_A^{V_{TX}}(r, t \to 0 | r_0)
= \begin{cases}
\dfrac{1}{4\pi r_0^2} \dfrac{\sqrt{a_{TX}^2 - (r - r_0)^2}}{2a_{TX}^2} & \text{if } r_0 - a_{TX} \leq r \leq r_0 + a_{TX} \\
0 & \text{otherwise}
\end{cases} \tag{11}
$$

In the equation, the release of the signaling molecule from the transmitting molecule is represented based on the volume.

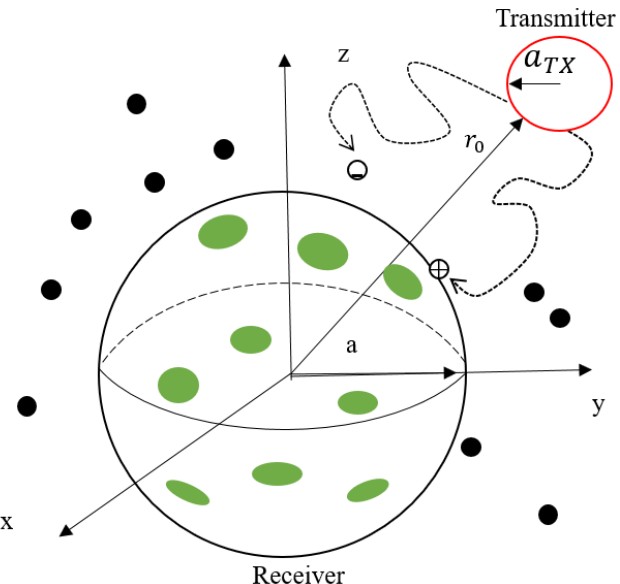

**Figure 5.** System model considering the volume of transmitter molecule.

### 3. Proposed Demodulation Method with Estimation of Ambient Environmental Parameters

*3.1. Conventional Threshold-Based Demodulation Method*

The demodulation by threshold is used to determine the reception sequence via comparison with threshold $\xi$, as expressed by

$$b_l^{RX} = \begin{cases} 1 & (\text{if } N_C^{\text{obs}}(t_l^{Threshold}|r_0) \geq \xi) \\ 0 & (\text{otherwise}) \end{cases} \tag{12}$$

where $b_l^{RX}$ denotes the first bit of the received bit sequence $b_{RX1}, b_{RX2}, \ldots, b_{RXl}, \ldots, b_{RXL}$. Furthermore, $t^{Threshold}$ denotes the time required to acquire $N_C^{obs}$ when demodulating with a threshold, which can be calculated as follows:

$$t_l^{\text{Threshold}} = (l - 1)T + t^{\text{Threshold}} \tag{13}$$

The sampling is based on the timing when the number of received molecules is maximum, which can be calculated using (8). Furthermore, the threshold value $\xi$ is set to a value that is 1/2 times the number of received molecules $N_C(t^{Threshold}|r_0)$.

$$\xi = \frac{1}{2} N_C(t^{\text{Threshold}}|r_0) \tag{14}$$

*3.2. Proposed MLSE Demodulation Method*

Maximum likelihood sequence estimation (MLSE)-based demodulation can determine the reception sequence as

$$\hat{b}^{\text{MLSE}} = \arg\max_{b} l(b) \tag{15}$$

where $b$ denotes the estimated bit sequence by MLSE, $b$ denotes the estimated candidate bit sequence, and $l(b)$ denotes the likelihood function for the estimated candidate bit sequence. We assume that the observation error of the number of received molecules follows a Gaussian distribution. The likelihood function is defined by the following equation.

$$l(b) = \prod_{k=1}^{K} p(N_C^{\text{obs}}|b)$$

$$= \prod_{k=1}^{K} \frac{1}{\sqrt{2\pi}\sigma} \exp\left[-\frac{\{N_{C,k}^{\text{obs}} - N_C^{\text{ISI}}(t = k\frac{T}{s_L}|r_0)\}^2}{2\sigma^2}\right] \tag{16}$$

We assume that the sequence length is $L$ and the symbol period $T$ is an oversampling of $st$ times, and the number of observation points is $K = s_L \cdot L$. The symbol $N_C^{\text{ISI}}$ denotes the number of received molecules and considers the intersymbol interference calculated based on (8). The estimation of channel parameters is expressed as

$$\hat{a}_0^{ML} = \arg\max_a l(a) \tag{17}$$

where $a_0^{ML}$ denotes the estimated value of $a$, and $l(a)$ denotes the likelihood function given by (16) for each estimated candidate. In this paper, we define the ambient parameter $a$ as the environmental temperature $T$ and distance $r$ between the transmitting and receiving molecules for each case, which can be estimated based on the above maximum likelihood estimation.

## 4. Performance Evaluation and Discussion

### 4.1. Computer Simulation Configuration

The results obtained by the simulation based on the CIR were compared with those obtained by the finite element method (FEM) in the system model of diffusive molecular communication. This confirmed that the system model proposed in the study can be used to simulate molecular communication based on the finite element method. As for the heterogeneous temperature distribution shown in Figure 4, the boundary between the regions is set at 0.25 μm from the transmitting molecule. The number of simulation trials was set to $10^6$ for the BER performance evaluation. Table 1 summarizes the simulation parameters.

**Table 1.** Channel parameters of diffusive molecular communication.

| | | |
|---|---|---|
| $N_A$ | Number of $A$ molecules | 5000 |
| $D_A$ | Diffusion coefficient of $A$ molecule | $5 \times 10^{-9} \text{ m}^2 \text{ s}^{-1}$ |
| $r_0$ | Transmitter and receiver distance | 1.0 μm |
| $a$ | Radius of receiver distance | 0.5 μm |
| $k_f$ | Forward reaction constant | $2.5 \times 10^{-14} \text{ m}^3 \text{ s}^{-1}$ |
| $k_b$ | Backward reaction constant | $2 \times 10^4 \text{ s}^{-1}$ |
| $k_d$ | Degradation reaction constant | $1 \times 10^4 \text{ s}^{-1}$ |
| $M$ | Number of recepters | 3000 |
| $r_s$ | Radius of recepter | 13.95 nm |
| $T$ | Temperature | 15, 25, 35 °C |
| $a_{TX}$ | Radius of Transmitter | 0, 0.10, 0.20, 0.30 μm |

To validate the simulation method based on the FEM in this study, Figure 6 demonstrates calculation examples obtained by simulation and theoretical results. It indicates that the simulation based on the finite element method is effective in diffusive molecule communication because theoretical values and simulation results are in good agreement. A simulation based on the finite element method was applied to a model in which the CIR cannot be easily derived.

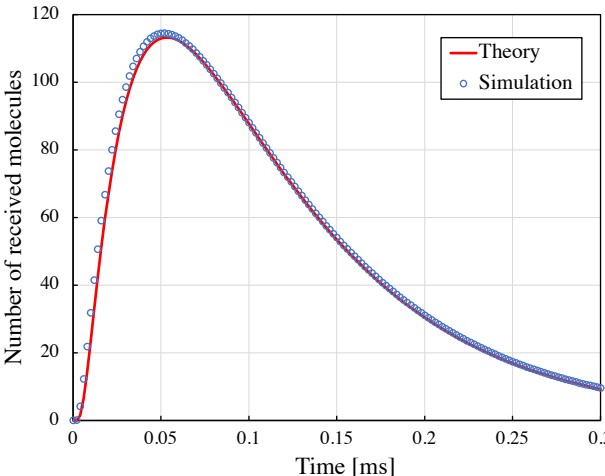

**Figure 6.** Calculation examples of number of molecules received in diffusive molecule communication.

As for the evaluation of the communication performance, we used a simulation to transmit transmission signals as sequence length $L$ and symbol period $T_p$. The communication characteristics of this setup were evaluated via the BER. Figure 7 shows the configuration of the BER performance evaluation with the parameters in Table 2. It is noted that the threshold-based method has a simple detection structure, whereas the MLSE method is based on the optimal detection based on the maximum likelihood estimation.

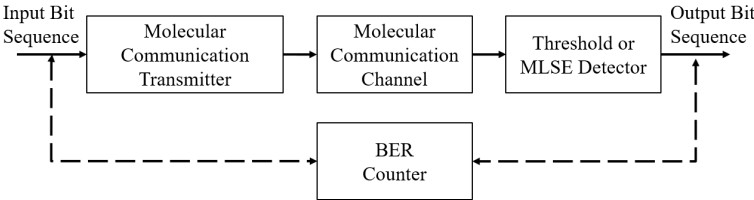

**Figure 7.** Simulation configuration for performance evaluation.

**Table 2.** BER simulation parameters.

| | | |
|---|---|---|
| $L_t$ | Training sequence length | 1 bit |
| $L$ | Data sequence length | 10 bits |
| $T_p$ | Symbol interval duration | 0.3 ms |

The signaling molecules released from the transmitting molecule reaches the surface of the receiving molecule by spreading the molecular communication channel. Therefore, inter-symbol interference is considered [23]. Additionally, additive white gaussian noise (AWGN) was added as an observation error when observing the number of received molecules. This is based on thermal noise in the amplifier, which is used for amplifying the acquired signal intensity when measuring light intensity with a photodetector using GFP as a receiver nanomachine [24].

### 4.2. Number of Received Molecules

Figure 8 shows the number of received molecules when the environmental temperature changes. As shown in Figure 8, the reaction becomes faster as the temperature is increased. This is because the value of the diffusion coefficient increases when temperature increases. Furthermore, it is observed that the rate stays the same after the temperature rise of over 30 °C, and the number of received molecules reaches the maximum at the temperature of 30 °C and then decreases. This is because there are two effects affected by the temperature. The number of received molecules are increased due to the temperature rise because the activity of the signaling molecules improves. On the other hand, the receptor of the receiver

molecule has a limitation to receive molecules in a certain time period, so the number of received molecules should be saturated when a large number of signaling molecules come to the receiver in a short time period.

Also, Figure 9 shows the number of received molecules when the temperature distribution is heterogeneous. At high temperature on the receiver side and low temperature on the transmitter side, the number of received molecules is increased compared to the setup with a homogeneous temperature of 35 °C. This is potentially due to the lower temperature on the transmitting molecule side, which reduces the reaction rate of the decomposition reaction. Conversely, when the temperature on the receiving molecule side was low, the number of received molecules decreased when compared to 30 °C due to the increased reaction rate of the decomposition reaction at higher temperature on the side of the transmitting molecule.

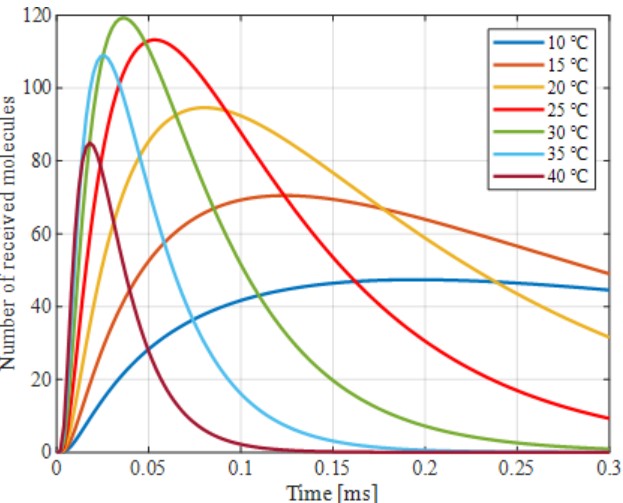

**Figure 8.** Number of received molecules in consideration of temperature changes.

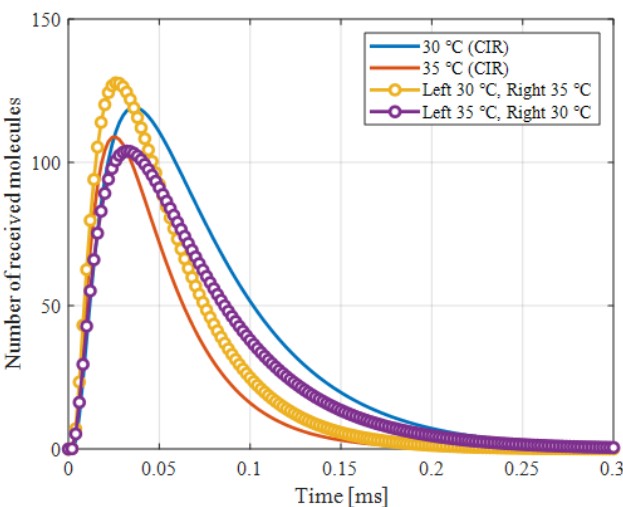

**Figure 9.** Number of received molecules when temperature distribution is heterogeneous.

The number of received molecules when the volume of the transmitted molecules changes is shown in Figure 10. The surface distance between the transmitting and receiving molecules was set as constant when the volume of the transmitting molecule was changed. The results confirmed that the number of received molecules tend to decrease due to increases in volume. It is considered that the number of molecules released from the position of $r = 1.0 \, \mu m$ decreased. This was realized by ensuring that the number of emitted molecules is the same as that in the case of a point source and by changing the volume by fixing the surface distance.

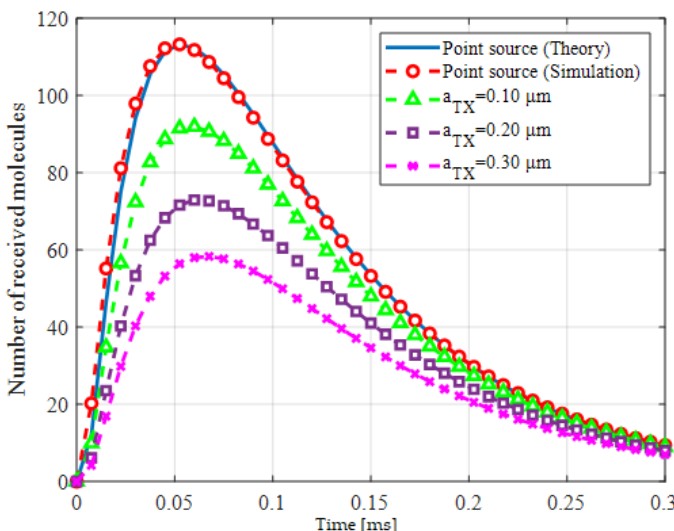

**Figure 10.** Number of received molecules considering the volume of transmitted molecules.

### 4.3. BER Improvement by MLSE Demodulation

To confirm the improvement in communication characteristics via the proposed MLSE method, the case of demodulation by the threshold value and the case of demodulation by the MLSE method were compared. The number of received molecules used in the BER simulation was calculated using Equation (8) by assuming that all channel parameters were known. Figure 11 shows the BER characteristics of the demodulation by the threshold and MLSE methods when the oversampling coefficients were varied as $s_L = 5, 15$, and $40$. The criterion for reliable communication was defined as a BER of $10^{-3}$. It can be shown that the demodulation by the MLSE method improves the communication characteristics when compared with demodulation by the threshold value. In particular, it is improved by $s_L = 40$.

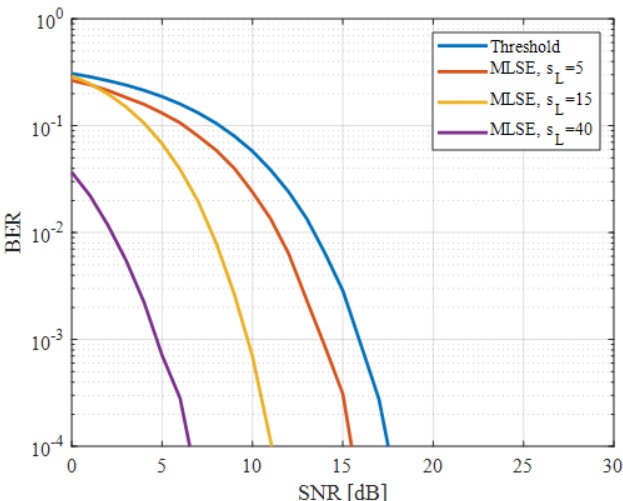

**Figure 11.** BER characteristics of demodulation method by the proposed MLSE method.

### 4.4. Evaluation of BER Characteristics with Respect to Temperature

The demodulation method shown in Table 3 is used in this study. Additionally, the default parameter shown in Table 3 assumes an ambient temperature of 25 °C and the parameter for the volume of the transmitter molecule is set as $r_0 = 1.0$ μm for demodulation.

**Table 3.** Summary of demodulation methods.

| | | Detection Method | Ambient Environment Parameters | Parameters in Detection |
|---|---|---|---|---|
| Conventional | Method A | Threshold | Known in advance | Exact parameters used (perfectly matched) |
| | Method B | MLSE | | |
| Proposed | Method C | Threshold | Unknown | Default parameters used (mismatch occurs) |
| | Method D | MLSE | | |
| | Method E | Threshold | | Ambient parameters estimated |
| | Method F | MLSE | | |

*4.5. Effect of Environmental Temperature on Communication Performance*

We evaluated the BER when the environmental temperature changes. The oversampling was set as $s_L = 15$. For the temperature estimation, the maximum likelihood estimation was adapted as expressed in (17). Figure 12a,b show the BER performances with respect to the temperature change, with and without the temperature information, respectively. As can be seen from in Figure 12a, whereas the BER performances for the threshold-based method (Method A) do not change in different temperature conditions, the BER performances for the MLSE method (Method B) change according to the ambient temperature. This means that the MLSE method can establish optimal detection even when the environmental temperature changes. Furthermore, from Figure 12b, Methods C and D cannot establish reliable communication because parameter mismatch occurs. On the other hand, Methods E and F can improve the BER performances based on the temperature estimation, which means the temperature is properly estimated. Based on the results in Figure 12, the signal-to-noise power ratio (SNR) that realizes a BER $\leq 10^{-3}$ is summarized in Table 4. As compared with Method A and B, the MLSE method (Method B) improves the SNR by 3.5 dB and 6.8 dB in the cases when the temperature is 25 °C and 35 °C, respectively. Furthermore, the performance degradation for the MLSE method is only 0.5 dB when the environmental temperature is necessary to be estimated, comparing with the results between Methods B and F at a temperature of 35 °C.

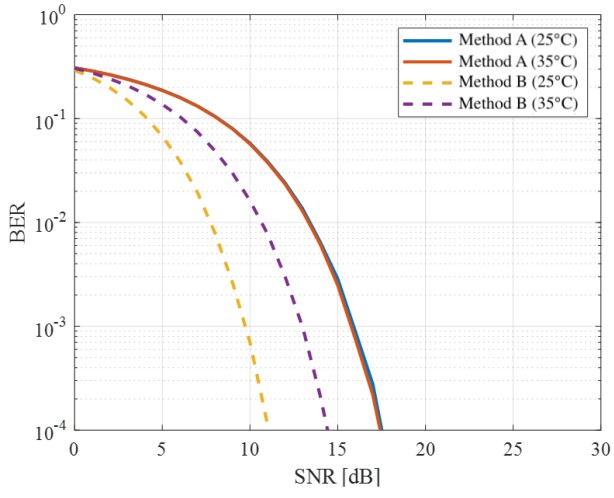

(a) Ambient temperature information is available.

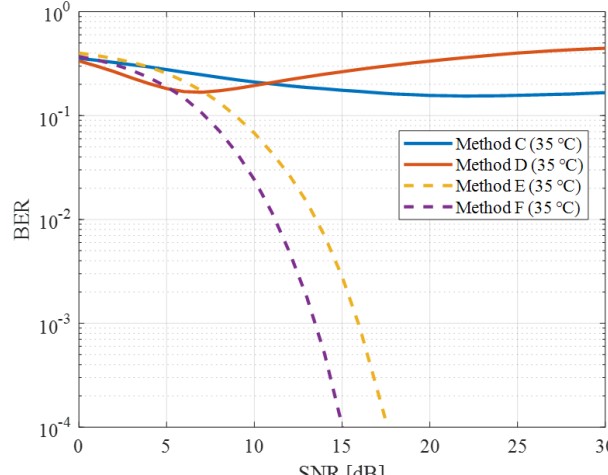

(b) Ambient temperature is unknown (fixed default parameters are used in Methods C and D, ambient parameters are estimated in Methods E and F).

**Figure 12.** BER characteristics with respect to temperature.

**Table 4.** Required SNR in homogeneous temperature distribution.

| Demodulation Method | Ambient Environment Parameters | Required SNR [dB] |
|---|---|---|
| Method A | 25 °C | 16.0 |
| Method A | 35 °C | 16.0 |
| Method B | 25 °C | 9.2 |
| Method B | 35 °C | 12.5 |
| Method C | 35 °C | - |
| Method D | 35 °C | - |
| Method E | 35 °C | 15.5 |
| Method F | 35 °C | 13.0 |

In addition to the evaluation in homogenous temperature environments, Figures 13 and 14 show the BER performances in heterogeneous temperature distributions. From Figure 13, the performance can be improved with the ambient parameter estimation for both threshold and MLSE methods. The same tendency is also observed in Figure 14. Based on the results in Figures 13 and 14, Table 5 summarizes the required SNR. The required SNR can be improved by more than 4 dB and 9 dB for the threshold method (comparison between Methods C and E) and MLSE method (comparison between Methods D and F), respectively, when the ambient parameter estimation is adapted under the heterogeneous temperature environment.

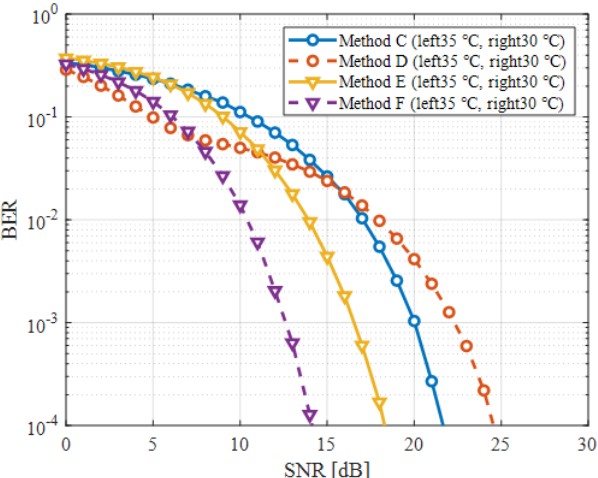

**Figure 13.** BER characteristics when the left region is 35 °C and the right region is 30 °C.

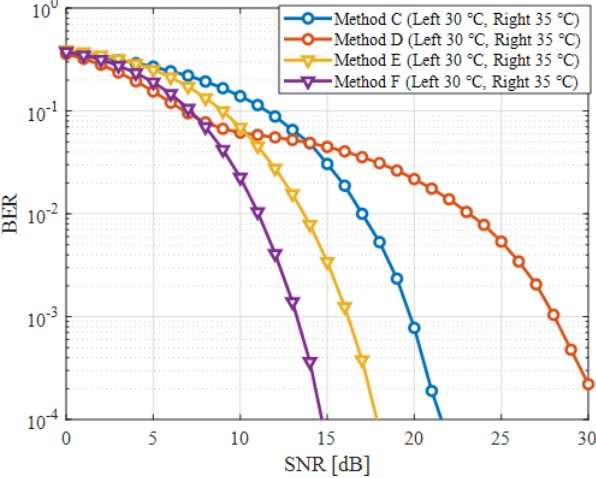

**Figure 14.** BER characteristics when the left region is 30 °C and the right region is 35 °C.

**Table 5.** Required SNR in heterogeneous temperature distribution.

| Demodulation Method | Ambient Environment Parameters | Required SNR [dB] |
|---|---|---|
| Method C | left 35 °C, right 30 °C | 20.2 |
| Method D | left 35 °C, right 30 °C | 21.9 |
| Method E | left 35 °C, right 30 °C | 16.1 |
| Method F | left 35 °C, right 30 °C | 12.1 |
| Method C | left 30 °C, right 35 °C | 22.5 |
| Method D | left 30 °C, right 35 °C | 27.7 |
| Method E | left 30 °C, right 35 °C | 15.8 |
| Method F | left 30 °C, right 35 °C | 12.8 |

*4.6. Effect of Transmitter Molecule Volume on Communication Performance*

We evaluated the BER for a model that considers the volume of the transmitting molecule. Here, the distance between the transmitting and receiving molecules was estimated based on the maximum likelihood estimation as expressed in (17). The oversampling was set to $s_L = 40$. Figure 15 shows the BER when the distance between transmitting and receiving molecules is estimated. Also, Table 6 summarizes the required SNR based on the results in Figure 15.

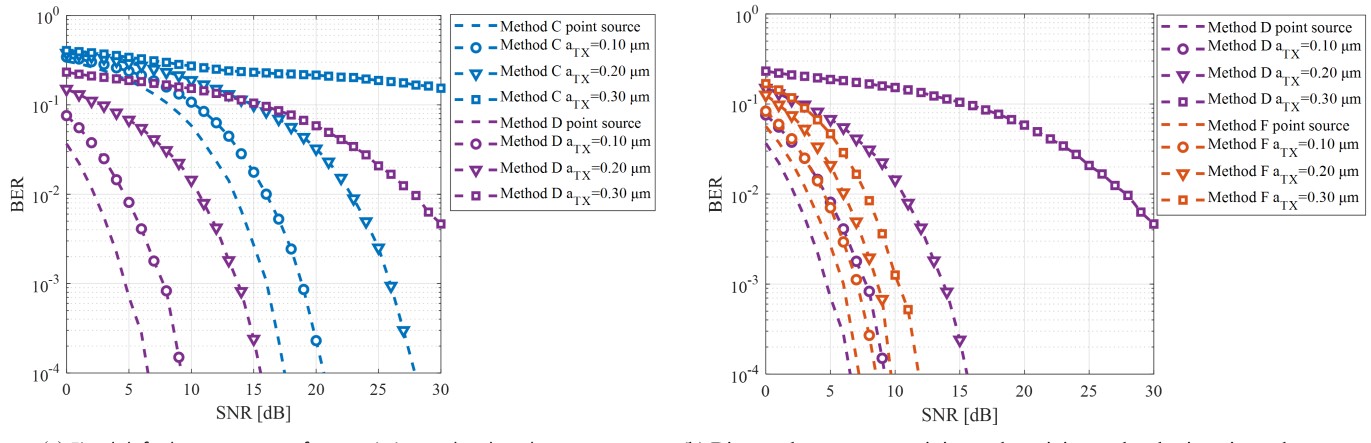

(a) Fixed default parameters of transmitting molecule volume are used (parameter mismatch occurs)

(b) Distance between transmitting and receiving molecules is estimated.

**Figure 15.** BER characteristics with respect to the volume of the transmitting molecule.

**Table 6.** Effect of the volume of transmitting molecule on Required SNR.

| Demodulation Method | Ambient Environment Parameters | Required SNR [dB] |
|---|---|---|
| Method C | point source | 16.0 |
| Method C | $a_{TX} = 0.10$ μm | 18.8 |
| Method C | $a_{TX} = 0.20$ μm | 25.9 |
| Method C | $a_{TX} = 0.30$ μm | - |
| Method D | point source | 4.6 |
| Method D | $a_{TX} = 0.10$ μm | 7.7 |
| Method D | $a_{TX} = 0.20$ μm | 13.8 |
| Method D | $a_{TX} = 0.30$ μm | - |
| Method F | point source | 6.0 |
| Method F | $a_{TX} = 0.10$ μm | 7.0 |
| Method F | $a_{TX} = 0.20$ μm | 8.6 |
| Method F | $a_{TX} = 0.30$ μm | 10.2 |

As presented in Table 6, by comparing Methods C and D, it is confirmed that the transmission improvements correspond to 11.4 dB when the transmitting molecule is a point source, 11.1 dB when $a_{TX} = 0.10$ μm, and 12.1 dB when $a_{TX} = 0.20$ μm. As shown

in Figure 15, communication characteristics also improve even when $a_{TX} = 0.30$ μm. Furthermore, in Table 6, a comparison of Methods D and F indicates that the transmission characteristics are also improved by 0.7 dB and 5.2 dB when $a_{TX} = 0.10$ μm and $a_{TX} = 0.20$ μm, respectively. The results indicated that the MLSE method can improve the communication characteristics even when actual volume of the transmitting molecule is assumed.

## 5. Conclusions

In this study, an analysis based on the finite element method was performed to study the proposed communication method and evaluate communication characteristics of the system model when the environmental temperature change and the transmitting molecule volume were assumed. The results indicated that, even in the case of heterogeneous temperature distribution, communication characteristics were improved by more than 9.0 dB by estimating temperature using the maximum likelihood estimation method. Additionally, the SNR that realizes a $BER \leq 10^{-3}$ was improved by approximately 11.1 dB by using the MLSE method under the assumption of actual volume of the transmitting molecule. A future study can focus on examining the communication characteristics under more realistic conditions such as continuous temperature variations with respect to time and location.

**Author Contributions:** Conceptualization, D.A.; methodology, J.K.; software, S.H.; validation, S.T.; formal analysis, S.T.; investigation, D.A. and G.F.; resources, D.A.; data curation, S.T.; writing—original draft preparation, S.T. and D.A.; writing—review and editing, J.K. and G.F.; visualization, S.H.; supervision, G.F. and D.A.; project administration, D.A.; funding acquisition, D.A. All authors have read and agreed to the published version of the manuscript.

**Funding:** This research and development work was supported by the MIC/SCOPE #JP225006004.

**Data Availability Statement:** Not applicable.

**Conflicts of Interest:** The authors declare no conflict of interest.

## Abbreviations

The following abbreviations are used in this manuscript:

| | |
|---|---|
| MC | Molecular Communication |
| DDS | Drug Delivery System |
| SIPONs | Superparamagnetic Iron Oxide Nanoparticles |
| MIMO | Multiple-Input Multiple-Output |
| CIR | Channel Impulse Response |
| BER | Bit Error Rate |
| GFP | Green Fluorescent Protein |
| OOK | On-Off Keying |
| MLSE | Maximum Likelihood Sequence Estimation |
| FEM | Finite Element Method |
| AWGN | Additive White Gaussian Noise |
| SNR | Signal-to-Noise power Ratio |

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
