# Peer review of "Ambient Environmental Parameter Estimation for Reliable Diffusive Molecular Communications"

_futureinternet, doi:10.3390/fi14110311_

Round 1

Reviewer 1 Report

The manuscript by Toriyama et al provides a systematic study on one type of molecular communications through diffusion process. Previously there have been similar studies that are of high interest to the research community, and the current manuscript aiming to simulate the molecular communications in the media of blood is also rather meaningful in the research field. The adopted dynamical  model is a diffusion process of one type of molecule from certain initial position, and various factors to influence the process are studied  by different methods for the numerical simulation. In my opinion, the presented results are sufficiently interesting to be published in the journal.   However, there a number of unclear points that should be clarified in a revised version.  They are listed as follows:
  1. For the diffusion model in Fig. 2 and the figures afterwards, it is not clear if the receivers, molecules B, distribute randomly or evenly on the illustrated sphere. Given the possible fluctuations of these receivers, could the connection of molecule A with them be affected to a certain extent? Maybe it is not possible to quantitatively predict the effect.
       2. The Bolztaman constant in Eq. (9) is not consistent with its explanation afterward.         3.  From where one can estimate the collision frequency A_k in Eq. (10)?         4.  In Fig. 6 the fitting of the simulation result with the theoretical result is obviously seen before the peak. Do they still fit well after crossing the peak since there is no red curve in this part?   After the minor revision regarding the above points, the manuscript can be accepted for publication. 

Reviewer 2 Report

The authors propose a model for the molecular communication environment using the finite element method, which takes into account the effect of temperature and volume of the transmitting source while using the MLSE method at the receiver. The paper is organized. The following are the reviewer's comments:

1- There are many English mistakes in the paper. Some sentence does not start with capital letters (like in P. 4 after (6)), and others are awkward like Line 119 "Fig. 8 shows the over time ...". Some repeated words and incomplete sentences also exist.

2- What is the difference between l(b) in (16) and l(a) in (18)? Both have the exact same definition.

3-   How many simulation runs are used and what is the confidence interval?

4- More explanation of Fig 8 is needed. It seems the rate stays the same after a certain temperature (30 degrees), why is that? Also, why does the number of received molecules reach the maximum with the same temperature (30 degrees) and then decrease? 

5- What is the implication of using the MLSE method compared with the threshold value method? Is it more complicated? 

6- In Line 173, how the distance between transmitting and receiving molecules can be estimated?

7- In Line 161, the explanation is not clear and does not refer to a specific figure. The authors should adequately explain each figure (12, 13, and 14) and then comment on the results.  

8- In line 166, the authors did not mention how the temperature can be estimated.
